# Deep Learning-Based Efficient Analysis for Encrypted Traffic

Xiaodan Yan [ID]

School of Cyber Science and Technology, Beihang University, Beijing 100191, China; yanxiaodan@buaa.edu.cn

**Featured Application: Analysis for encrypted traffic.**

**Abstract:** To safeguard user privacy, critical Internet traffic is often transmitted using encryption. While encryption is crucial for protecting sensitive information, it poses challenges for traffic identification and poses hidden dangers to network security. As a result, the precise classification of encrypted network traffic has become a crucial problem in network security. In light of this, our paper proposes an encrypted traffic identification method based on the C-LSTM model for encrypted traffic recognition by leveraging the power of deep learning. This method can effectively extract spatial and temporal features from encrypted traffic, enabling accurate identification of traffic types. Through rigorous testing and evaluation, our system has achieved an impressive accuracy rate of 96.4% on the widely used ISCXVPN2016 dataset. This achievement demonstrates the effectiveness and reliability of our method in accurately classifying encrypted network traffic. By addressing the challenges posed by encrypted traffic identification, our research contributes to enhancing network security and privacy protection.

**Keywords:** data security; encrypted traffic identification; distributed learning; artificial intelligence security

## 1. Introduction

The popularity of applications such as smart homes and smart healthcare systems has led to an increasing volume of massive data generated by heterogeneous IoT devices. Machine learning-based traffic identification technology can successfully detect traffic anomalies on a network, thus revealing unknown network attacks and providing a basis for deploying defense measures. With the spread of encryption technologies such as HTTPS (Hypertext Transfer Protocol Secure) and VPN (Virtual Private Network), much critical traffic on the Internet is transmitted in encrypted form. According to Google statistics [1], 95% of the sites visited by its users as of January 2021 are encrypted with HTTPS. Encryption technology to preserve user privacy also creates further challenges in traffic identification tasks and has evolved swiftly in recent years. Deep learning technology provides a solution for encrypted traffic identification [2]. The neural network is trained using a large-scale dataset and can be automated to extract features from the flow, enabling the implementation of an end-to-end encrypted traffic identification method. This method can be utilized for detecting anomalous traffic and managing service quality. Malicious traffic disguises itself as normal traffic, making it challenging to effectively identify encrypted traffic without complex feature engineering.

The contemporary traffic identification method is predominantly aimed at traffic transmitted in the form of plaintext. However, encrypted traffic is traffic generated after the content that needs to be transmitted on the network is encrypted using an encryption algorithm [3–5]. Traditional traffic identification methods include identification methods based on port matching and deep packet inspection [6], which are more effective for nonencrypted traffic identification tasks. For encrypted traffic identification, it is necessary to rely on statistical-based machine learning methods, and fast deep learning approaches have been developed in recent years. The identification method based on port matching is to identify the data packet through the service port number specified by the Internet

Assigned Numbers Authority (IANA), which provides the ports corresponding to some basic recognized services. The first 1024 ports are reserved as privileged ports, and the ports after 1024 are allocated by the system as dynamic ports. The identification method based on port matching only utilizes the information provided by the header field in the transport layer, which is highly efficient in the identification process and could deal with a single type of application traffic in the early Internet but cannot adapt to the current network environment. Due to the widespread use of peer-to-peer (P2P) and network address port translation (NAPT) technologies and the existence of virus programs using port masquerading technology, the data packet ports in network traffic do not always correspond to services. Numerous traffic types are identified with low accuracy.

The identification method based on DPI performs pattern matching by searching for information in the entire IP data packet, including the header fields of the network layer, transport layer, application layer, and application layer payload [7–9]. Pattern matching algorithms include simple string matching and pattern matching based on hash functions. For encrypted traffic in this method, the data packet load is theoretically randomly distributed. Although certain patterns and statistical laws depend on implementing specific encryption algorithms, they cannot be used in DPI methods based on simple strings. Pattern matching with hash operations is used for detection and identification.

Current network applications mostly use encryption methods such as HTTPS, SSH, and VPN to encrypt the data that need to be transmitted in the public network [10]. During the encryption process, only the initial session establishment process will use plaintext communication to complete certificate verification and key generation. In the actual communication process, the payload in the data packet will be encrypted using a symmetric cipher and deep packet inspection technology cannot be used to identify the traffic type corresponding to the data packet, but it can be further analyzed. The identification task of encrypted traffic can be conducted to some extent by using machine learning methods to adapt a series of statistical features, or spatial and temporal features of a single data packet, during the transmission of a data stream.

With the development of deep learning technology, convolutional neural networks (CNNs) [11] and recurrent neural networks (RNNs) [12] have provided numerous mature solutions for computer vision and natural language processing. Using deep learning techniques, features in traffic can be automatically extracted without feature engineering through multiple nonlinear fully connected layers of artificial neural networks with excellent generalization capabilities.

This study proposes a deep learning-based solution for encrypted traffic identification and clarifies the subject background and purpose of the proposed deep learning-based encrypted traffic identification system. The aim of this research is to design and implement a deep learning-based system for accurately identifying encrypted traffic. Initially, we construct a CNN model to learn spatial characteristics from the training and then evaluate its performance in identifying encrypted traffic using the testing dataset. Next, we introduce an enhanced LSTM model that utilizes the feature generated by the convolutional layer of our one-dimensional CNN model as input. This allows us to utilize high-level spatial features and extract temporal characteristics among them. These modifications enable us to develop a comprehensive model for accurately identifying encrypted traffic. This system effectively accomplishes end-to-end encrypted traffic identification by automatically extracting temporal features from packets. The contributions of this work can be summarized as follows.

(1) This paper aims to develop a deep learning-based encryption traffic identification system that automatically extracts spatial and temporal features from data packets to achieve end-to-end encryption traffic identification.

(2) The overall framework of the encrypted traffic identification system based on deep learning is proposed, and the relevant details of the data flow are analyzed. We also optimize the efficiency of storage space and running time for data preprocessing.

(3) The implementation of a comprehensive model for identifying encrypted traffic was conducted, and a quantitative comparison with traditional detection was carried out This study validates the proposed C-LSTM model for extracting spatial and temporal features from data packets.

This paper is organized as follows: We illustrate the introduction, contributions, and related works in Section 1. Section 2 illustrates the preliminaries of the work. The system framework is described in Section 3. Section 4 describes the proposed scheme and performance analysis, respectively. Finally, the paper is concluded in Section 5.

## 2. Related Work

The port matching identification method only uses the header field of the transport layer, which is efficient but not adaptable to current network environments. This is because recent P2P applications use random port policies to avoid detection and blocking [13]. Strict firewalls prohibit access to unknown ports by default, but viruses can exploit port-masquerading techniques to hack into systems. For example, DNS tunnel Trojans hide information using the domain name returned by port 53 during DNS queries [14,15], while WannaCry spreads through SMB protocol port 445 [16]. Based solely on port matching, computer systems can be compromised.

The Knuth–Morris–Pratt algorithm, which improves the right shift rule of the brute-force algorithm, uses the previously saved pattern-matching information to move the detected string farther to the right, thereby improving the time efficiency of the algorithm [17]. For functions commonly used in the web field, such as URL filtering, the BM (Boyer–Moore) algorithm is also widely used. Compared with the KMP algorithm, it increases the distance of each right shift and performs actual tasks. It is relatively helpful and simple to implement. It can handle large-scale URL filtering tasks and can also perform deep packet inspection on the data packets in the traffic.

Recently, there have been attempts to apply deep learning in traffic identification. Wang et al. [18] proposed using a simple one-dimensional convolutional neural network (1D-CNN) for traffic identification, which is the first time end-to-end encrypted traffic detection has been realized. Ramakrishnan et al. [19] applied standard RNN, long short-term memory (LSTM), and gated recurrent units (GRU) in recurrent neural networks to network traffic prediction, and their accuracy was better than that of the traditional method.

To continue to enhance the recognition accuracy of the model for the convolutional neural network, the structure of the network needs to be widened and deepened, which will confront the problem of increasing the amount of training or overfitting. For networks, continuing to increase the accuracy of the model with the same input sequence also requires increasing the depth of the network, which leads to the problem of vanishing or exploding gradients, making the training process of the model impossible to continue. Chawla et al. [20] combined a convolutional neural network with a recurrent neural network and applied it to a host-based intrusion detection system, achieving an anomaly-based intrusion detection system with high computational performance. In the field of traffic anomaly detection, Xu et al. [21] proposed the HBiRNN method, which combined two-dimensional RNN and LSTM to realize a hierarchical learning method of spatial and temporal features for session flows in network traffic. This approach has a higher detection rate and a lower false alarm rate.

The traditional methods for traffic identification primarily focus on plaintext traffic, while encrypted traffic is generated by encrypting the transmitted content using encryption algorithms. Consequently, these conventional methods exhibit limited effectiveness when dealing with encrypted traffic. Matching and deep packet inspection are techniques that perform well in identifying unencrypted traffic. To effectively identify encrypted traffic, machine learning, and deep learning techniques are now employed. These approaches have the ability to capture patterns and characteristics of encrypted traffic more effectively without requiring excessive manual feature engineering. The advantage of such methods

lies in their capability to automatically detect algorithms and traffic patterns, thereby enhancing the accuracy of identifying encrypted traffic.

## 3. System Framework of Encrypted Traffic Identification

The input to the system is the raw traffic dataset. All steps of the data preprocessing module, training, and test sets suitable for deep learning model processing are generated. The training set contains packet information from traffic and its corresponding labels, which are used for training deep learning models. Spatiotemporal features are automatically extracted by concatenating CNN models and LSTM models. After completing the optimization and verification of the training set, the trained deep learning model is obtained. Finally, it is evaluated through the test set. The model output is transformed by the softmax layer, the output label is obtained, it is compared with the real label, and the model's accuracy and additional indicators are calculated as a reference. The overall system framework is illustrated in Figure 1.

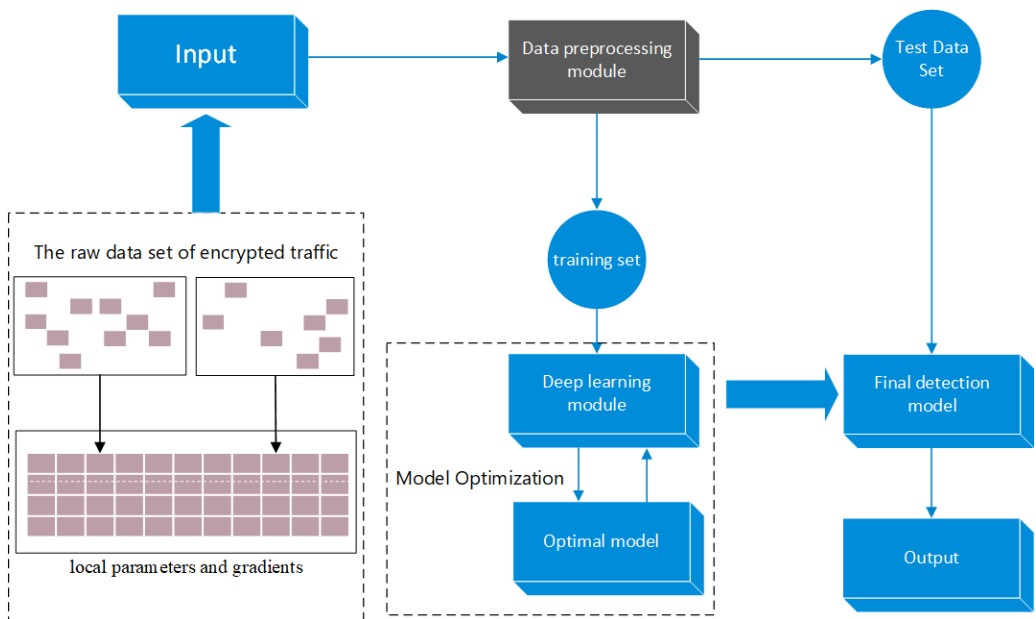

**Figure 1.** The framework of an encrypted traffic identification system based on CNN.

### 3.1. Preprocessing Methods for Encrypted Traffic Datasets

This paper uses the ISCXVPN2016 dataset [22] published by researchers at the University of New Brunswick, which contains a set of PCAP (packet capture) files, including six different types of application traffic and their corresponding OpenVPN. In total, there are 12 types of encrypted traffic.

The traffic types and specific network applications or protocols included in this dataset are listed in Table 1 [18]. Protocols such as SMTPS use SSL/TLS protocols at the transport layer for encryption. Applications such as BitTorrent and Skype use customized encryption methods at the application layer. In addition, there are six traffic types that use OpenVPN to encrypt the aforementioned application traffic at the network layer, which are implemented based on the SSL/TLS components provided by the OpenSSL library.

The content of the PCAP file provided by the dataset is a file organized according to the standards specified by the application programming interface (API) of the same name. In the Windows system, it corresponds to the WinPcap and Npcap libraries, which are used to monitor and capture the traffic in the network in real time. Each data packet contains the header fields and payloads of layers 2–7 in the OSI model. The exported PCAP file structure is shown in Figure 2.

**Table 1.** ISCXVPN2016 dataset traffic content [18].

| Traffic Type | Application or Protocol |
| --- | --- |
| Email (VPN-Email) | SMTPS, POP3S, IMAPS |
| Chat (VPN-Chat) | ICQ, AIM, |
| Streaming (VPN-Streaming) | Vimeo, YouTube |
| File Transfer (VPN-File Transfer) | Skype, FTPS, SFTP |
| VoIP (VPN-VoIP) | Facebook, Skype, Hangouts |
| P2P (VPN-P2P) | uTorrent, Transmission |

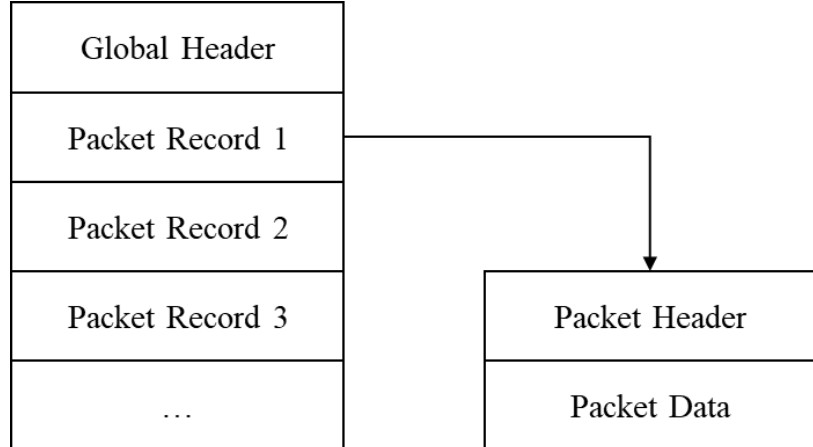

**Figure 2.** PCAP file structure.

In PCAP files, the initial 24 bytes represent the global header, which provides essential details about the file version, timestamp, and the network's data link. Following the global header, there are multiple packet records. The packet header contains metadata pertaining to each packet, such as the capture time and packet length. Packet data refer to the actual packet content.

*3.2. Data Cleaning for Encrypted Traffic*

Data collected in real-world network environments often include packets that are irrelevant to a particular application. If there is an excessive presence of such irrelevant packets within a particular traffic type, and their statistical properties significantly deviate from those of normal packets, it may hinder the convergence of the deep learning model and hinder the accurate identification of the corresponding traffic type. After analyzing the pcap files in the data set using the Wireshark tool, it was found that the collected data consisted of packets from ARP, DNS, NBNS, and LLMNR protocols. These protocols are used in networked environments to establish routing and resolve domain names and physical addresses. However, they do not provide information pertaining to specific application packet characteristics. Therefore, filtering operations should be performed during the traffic cleaning procedure. The traffic associated with the TCP protocol also encompasses packets for the initiation and termination of TCP connections. These data packets can enable the normal establishment and termination of the link and complete congestion control on the link. The integrity of the payload is checked but it does not carry information related to specific applications, which is not helpful for the deep learning model to extract traffic characteristics. After filtering, the size of the dataset can be reduced without losing information related to the recognition task. It can speed up the convergence rate of the model loss function during the training process. The algorithm for the encrypted traffic cleaning step is shown in Algorithm 1.

| **Algorithm 1:** Traffic Cleaning Algorithm |
| --- |
| **Input: Structured Packet Object,packet**<br>**Output: Boolean,Determines whether the packet object needs to be filtered** |
| 1　　**function** PACKET-FILTER (packet)<br>2　　**if** packet.protocol in (ARP, DNS, NBNS, LLMNR) **then**<br>3　　**return** true<br>4　　　　**else if** packet.protocol = TCP **then**<br>5　　　　**if** packet.payload.length = 0 or packet.flag & 0x07! = 0 **then**<br>6　　**return** true<br>7　　**return** false |

### 3.3. Anonymization of Encrypted Traffic

Since the traffic monitoring tools used in this dataset are Wireshark and TCP dump, both of which collect data packets on the data link layer, the output PCAP file contains all the packet headers above the data link layer, including the MAC address and IP address of the network adapters in the computers of both communicating parties. Considering that this information is only related to the actual network devices, such as the network adapter and router configurations, specific network applications usually do not configure these properties, and the dataset authors used the same devices when collecting the same traffic type. As a result, these attributes directly determine the type of data packets and the neural network tends to fit these features, resulting in the problem of high recognition accuracy but poor generalization ability. Therefore, to prevent the neural network from fitting these features, it is necessary to anonymize the data link layer and the network layer.

For the transport layer, since many network applications develop their own communication modules based on the socket protocol, information related to specific types of applications will be in the transport layer header, such as the TCP source port number and the destination port number, which should be reserved. In traffic anonymization, for performance reasons, the data link layer and network layer headers of the data packets are directly deleted, and only the transport layer header and the upper layer payload are retained. The traffic anonymization algorithm is as Algorithm 2.

| **Algorithm 2:** Traffic Anonymization Algorithm |
| --- |
| **Input:Packet Object, packet**<br>**Output:Anonymized Packet Object, packet** |
| 1　　**function** PACKET-ANONYMIZATION(packet)<br>2　　　　packet.remove_head(L2, L3)<br>3　　　　**return** packet |

### 3.4. Data Padding and Truncation for Encrypted Traffic

The protocols at the transport layer include TCP and UDP protocols. The length of the header is 20 bytes and 8 bytes, respectively. The payload starts from the 21st and 9th bytes of the packet. The corresponding input vectors of the same part cannot be aligned in dimensionality, which affects the recognition effect of the model for different protocols. Therefore, to align the dimensions of the feature vectors corresponding to the application layer loads of the two protocols so that the convolutional neural network can extract the correct spatial features, the UDP header fields can be processed with zero padding, as shown in Figure 3. As shown, 12 bytes of all-zero data are filled after the header field of the UDP data packet so that the sum of the header length of the original data packet and the zero-filled content is 20 bytes, which is the same length as the header field of the TCP protocol.

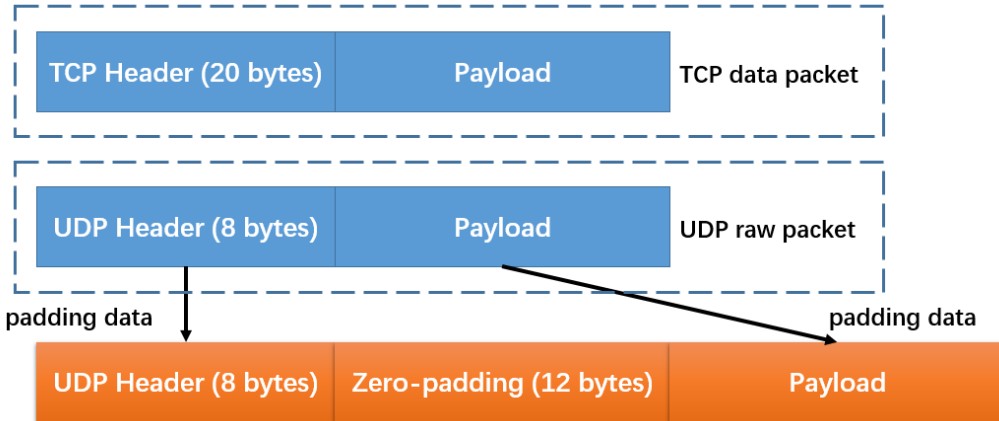

**Figure 3.** The framework of the encrypted traffic identification system based on CNN.

The maximum length of an IP datagram should be 1500 bytes. After removing the network layer header in the traffic anonymization step, the packet length will decrease by 20 bytes, so the total length of the packet should be 1480 bytes for only the transport layer and all layers above it. Finally, in implementing the padding and truncation steps, the data packets are padded with insufficient length and the data packets that are too long are truncated so that the length of all data packets is 1480 bytes. The pseudocode is shown in Algorithm 3.

---

**Algorithm 3:** Data Padding and Truncation Algorithms

---

**Input:Packet Object, packet**
**Output:PADDING Packet Object, packet**

---

1     **function** PADDING-AND-TRUNCATION(packet)
2       **if** packet.protocol = UDP **then**
3         packet←packet.head/0x00 * 12/packet.payload
4       **if** packet.length < 1480 **then**
5         packet←packet/0x00 * (1480 − packet.length)
6       **else if** packet.length > 1480 **then**
7         packet←packet[1:1480]
8       **return** packet

---

### 3.5. Anonymization of Encrypted Traffic

Normalization makes the input vector conform to the deep learning framework standard and speeds up the optimization process of neural network training. As is known, a byte has 8 bits that can represent 256 values (0 to 255). The input vector should be normalized. Package $P$ can be thought of as a byte array of length $n$ of 1480: $(p_1, p_2, \ldots, p_n)$. We convert the data to a float, as a float is more suitable for CPU or GPU calculations. Through normalization, the range of feature values of different dimensions is adjusted to a similar range, and a large learning rate can be uniformly used to accelerate learning. The value range of element $p_i$ is $[0, 2^8)$. When we divide the data by 255, we obtain a value in the range of 0 to 1. We express the normalization operation as the following formula.

$$x_i = \frac{p_i}{p_{max} - p_{min}} = \frac{p_i}{255} \tag{1}$$

The normalized vector $X = x_1, x_2, \ldots, x_n \in [0, 1]^n$ is obtained, and the pseudocode of the normalization step is shown in Algorithm 4.

| **Algorithm 4:** Traffic Normalization Algorithm |
|---|
| **Input:Packet object, packet** <br> **Output: NormalizedPacket Object, packet** |
| 1　　**function** PACKET-NORMALIZATION(packet) <br> 2　　　　**for** $i$ = 1 to 1480 do <br> 3　　　　　　packet[$i$]←packet[$i$]/255 <br> 4　　　　**return** packet |

After preprocessing all the packets in the dataset, we obtain an array of input vectors suitable for the deep learning framework. Next, we construct a suitable deep learning model to complete traffic feature extraction. The algorithm reads each PCAP file in the dataset online. Additionally, the packets are read into memory individually, and the protocol of each layer is parsed. Normalization conforms the input vectors to the standard of deep learning frameworks and speeds up the optimization process of neural network training. The input vector should be normalized.

*3.6. Split Training Sets, Validation Sets, and Test Sets*

To mitigate the issues of overfitting and underfitting during the training process, we partition the data into three subsets: training set, validation set, and test set. The purpose of the training set is to facilitate parameter and weight optimization for our model, while the test set serves as a means to assess its final performance and generalization capability. As for the validation set, it plays a crucial role in providing an unbiased evaluation of how well our model fits tuning. We stratify the preprocessed data by sample type, sampling packets from each PCAP file in proportion to the corresponding traffic type, with the goal of maximizing sample coverage. We allocated 20 percent of the data as the test set, while the remaining 80 percent was split into a training and validation set. For training purposes, we employed 10-fold cross-validation. This involves dividing the training set into ten equal parts, where nine parts are used for training in each round, and one part is used for validation. Following each validation round, we fine-tuned the model's structure and layer hyperparameters. The final optimized model was determined by selecting the best hyperparameters, after which a final evaluation was conducted on the test set, yielding various performance metrics.

**4. Encrypted Traffic Identification Method Based on Deep Learning**

For encrypted traffic identification tasks, a single data packet in the collected network traffic can be regarded as a byte array. This array includes the header field of the packet and the payload, which conforms to the encoding rules of a particular network protocol or application. It is suitable for extracting spatial features using a one-dimensional CNN model. In this paper, we first build a one-dimensional CNN model to train and test the preprocessed dataset to verify the feasibility of the deep learning model for the task of identifying encrypted traffic.

The one-dimensional CNN model used in this paper contains two convolutional layers, one maximum pooling layer (Max-pooling Layer), and five fully connected layers (Fully Connected Layer or Linear Layer). The convolution transformation of the convolution layer can extract the features of local adjacent bytes in the array, which is similar to the n-gram model in natural language processing. Then, this layer abstracts the rules of the protocol or encoding and stores multiple local features through a set of feature vectors. A schematic diagram of the structure of the one-dimensional CNN model used in this paper is shown in Figure 4. In the process of building the actual deep neural network, the dropout operation is introduced into the fully connected network. When the parameters are updated by the backpropagation algorithm, the iteration of some parameters can be canceled with a certain probability to avoid overfitting. In addition, ReLU is chosen as the nonlinear activation function at the output of the convolutional layer and the fully connected layer of the last layer.

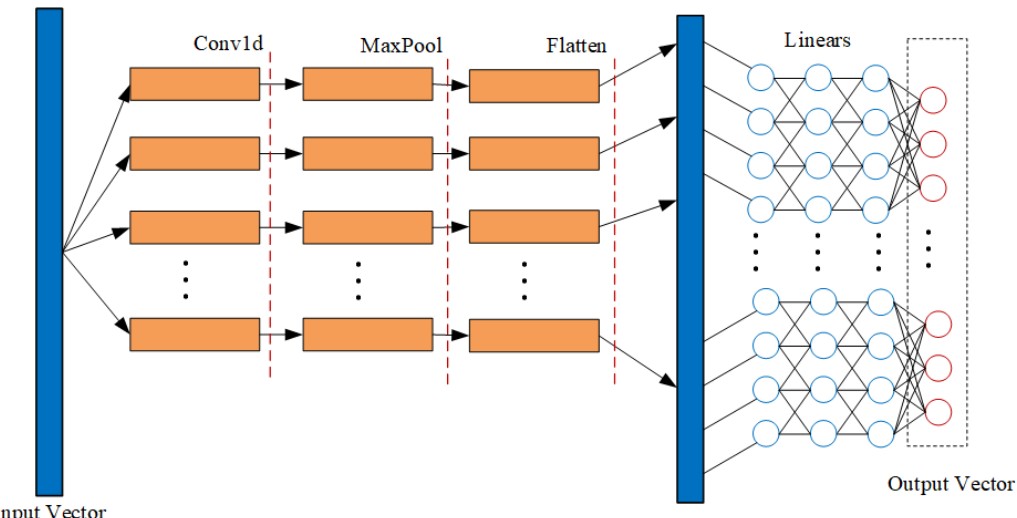

**Figure 4.** One-dimensional CNN model structure diagram.

The dropout rate of the fully connected layer is set to 0.05 in the training process to prevent overfitting the model. We chose the cross-entropy function as the loss function and used the Adam optimizer, which has an adaptive learning rate, for iteration. The batch size of the data is 32; that is, all 32 vectors in the batch are passed through the network in one iteration. The output of the cross-entropy loss function was then calculated for the whole batch, and the parameters in the network were updated accordingly. A total of three iteration rounds were performed on the complete training set.

In Section 3, we introduce the overall framework of the encrypted traffic identification system based on deep learning, providing an in-depth analysis of the role of each module within the system by examining data flow details. We then derive the details of the data preprocessing steps, offering a precise description in the form of pseudocode. Additionally, we present an analysis of the adopted open-source encrypted traffic dataset in this section, exploring the data structures and various traffic types within it. Moving on to the deep learning model, we analyze the structure and principle of the existing model, gradually building the detection model. Finally, we utilize softmax as the last layer classifier for traffic identification, completing the construction of the entire encrypted traffic identification system.

We perform statistical analysis on different traffic types in the ISCXVPN2016 dataset. Consider the Email type, which corresponds to four PCAP files. We first sum up the file sizes to count the quantity of data we need to process and provide a reference for later preprocessing work. We count all available packets by filtering them according to the traffic-cleaning steps described in Section 3. Figure 5 shows the traffic statistics for all 12 types.

The one-dimensional CNN model used in this paper contains two convolutional layers, one maximum pooling layer, and five fully connected layers. ReLU is added to the two convolutional layers and five fully connected layers as a nonlinear activation function, and the specific hyperparameters are shown in Table 2.

On the basis of CNN, the LSTM model is further added to extract the temporal features of the feature vector sequence. The CNN and LSTM (C-LSTM) model employed in this study shares similarities with the aforementioned CNN, comprising two convolutional layers and a max-pooling layer. Then, we utilize the output of the max-pooling layer as the input for the subsequent LSTM layer, which has a hidden layer size of 50. The resulting output is flattened and fed into a 3-layer fully connected network for fitting, with the hyperparameters detailed in Table 3.

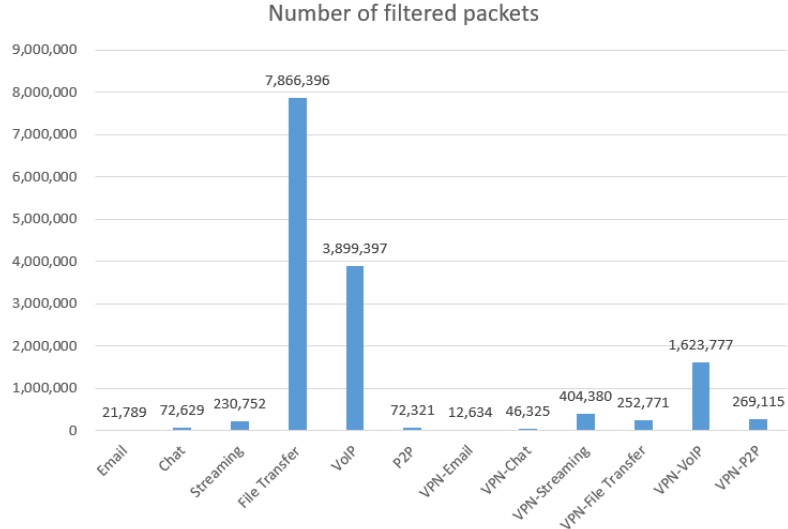

**Figure 5.** Dataset Statistics.

**Table 2.** Temperature and wildlife count in the three areas covered by the study.

| Layer | Operation | Kernel Size | Step-Size | Input Size | Output Size |
|---|---|---|---|---|---|
| 1 | Conv1d + ReLU | 5 | 3 | $1 \times 1480$ | $50 \times 492$ |
| 2 | Conv1d + ReLU | 4 | 3 | $50 \times 492$ | $50 \times 163$ |
| 3 | MaxPool1d | 2 | 2 | $50 \times 163$ | $50 \times 81$ |
| 4 | Linear + Dropout + ReLU | - | - | $1 \times 4050$ | $1 \times 400$ |
| 5 | Linear + Dropout + ReLU | - | - | $1 \times 400$ | $1 \times 200$ |
| 6 | Linear + Dropout + ReLU | - | - | $1 \times 200$ | $1 \times 100$ |
| 7 | Linear + Dropout + ReLU | - | - | $1 \times 100$ | $1 \times 50$ |
| 8 | Linear | - | - | $1 \times 50$ | $1 \times 12$ |

**Table 3.** Hyperparameters for the C-LSTM model.

| Layer | Operation | Kernel Size | Step-Size | Input Size | Output Size |
|---|---|---|---|---|---|
| 1 | Conv1d + ReLU | 5 | 3 | $1 \times 1480$ | $50 \times 492$ |
| 2 | Conv1d + ReLU | 4 | 3 | $50 \times 492$ | $50 \times 163$ |
| 3 | MaxPool1d | 3 | 3 | $50 \times 163$ | $50 \times 81$ |
| 4 | LSTM | - | - | $50 \times 81$ | $50 \times 50$ |
| 5 | Linear + Dropout + ReLU | - | - | $1 \times 2500$ | $1 \times 500$ |
| 6 | Linear + Dropout + ReLU | - | - | $1 \times 500$ | $1 \times 50$ |
| 7 | Linear | - | - | $1 \times 50$ | $1 \times 12$ |

First, we evaluated the 1D CNN model constructed and trained in this paper, which achieves 94.6 percent accuracy and can efficiently identify TLS and VPN-encrypted traffic. The accuracy rates are depicted in Figure 6, whereas the recall rates are portrayed in Figure 7.

In the context of precision and recall, the one-dimensional CNN model developed in this paper outperforms traditional machine learning algorithms, such as the C4.5 algorithm and the KNN algorithm, both of which rely on manually selected data features. The model constructed in this paper is compared with other methods for traffic identification based on the ISCXVPN2016 dataset. The precision and recall rates of each model are shown in Figure 8.

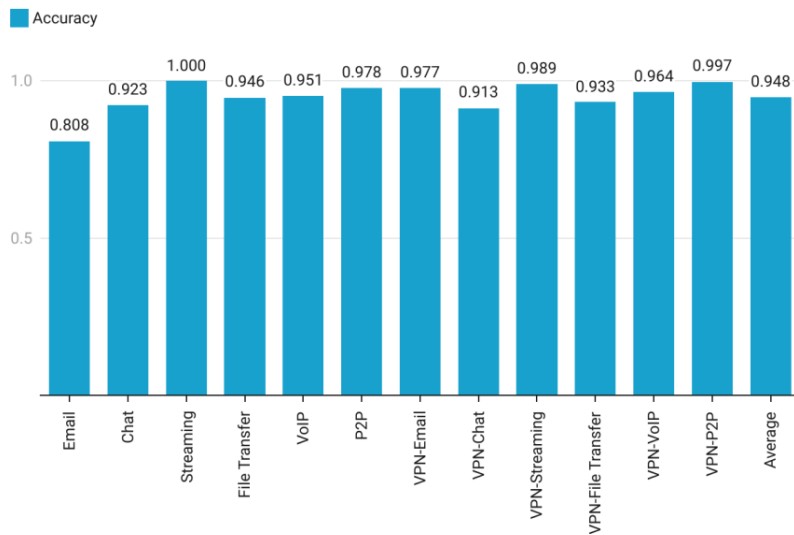

**Figure 6.** Accuracy rate of the one-dimensional CNN model on various traffic flow types.

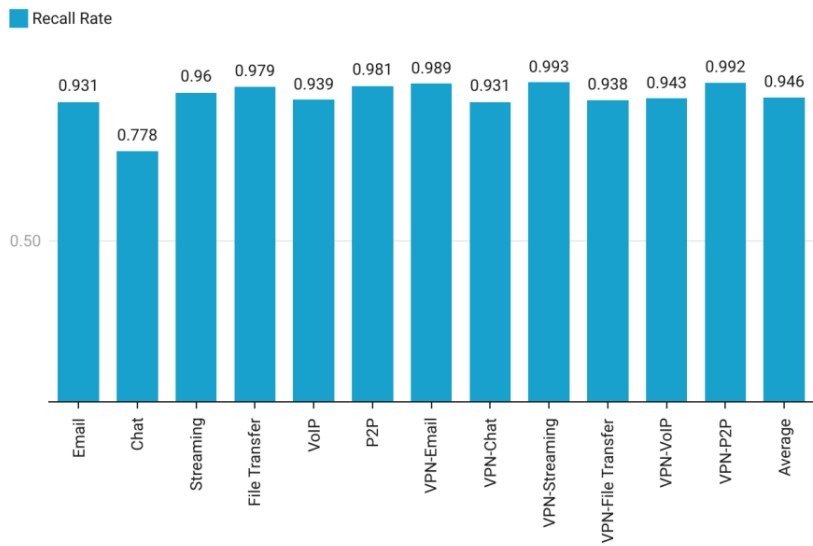

**Figure 7.** Recall rate of the one-dimensional CNN model on various traffic flow types.

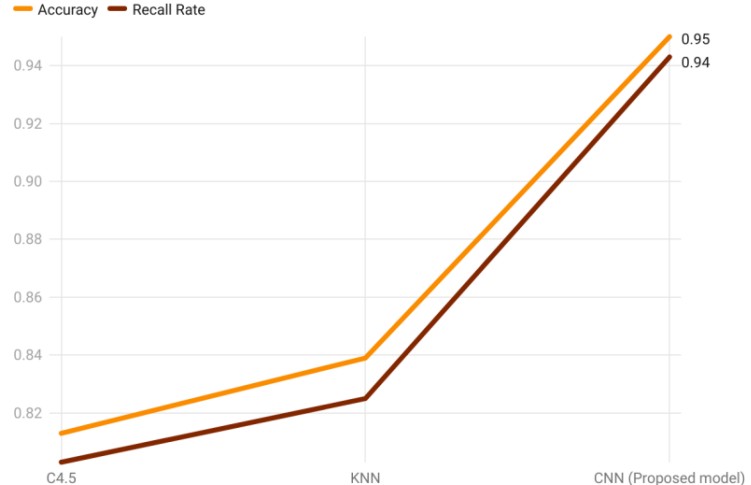

**Figure 8.** Accuracy and recall rate comparison of models.

The one-dimensional CNN-based encrypted traffic identification model effectively detects TLS and VPN encrypted traffic. We compared the predicted labels with the true labels obtained from the test set packets and the resulting matrix is shown in Figure 9. Each row represents the actual traffic type, while each column represents the predicted traffic type based on the one-dimensional CNN model. It is evident that the model demonstrates excellent performance across most traffic types.

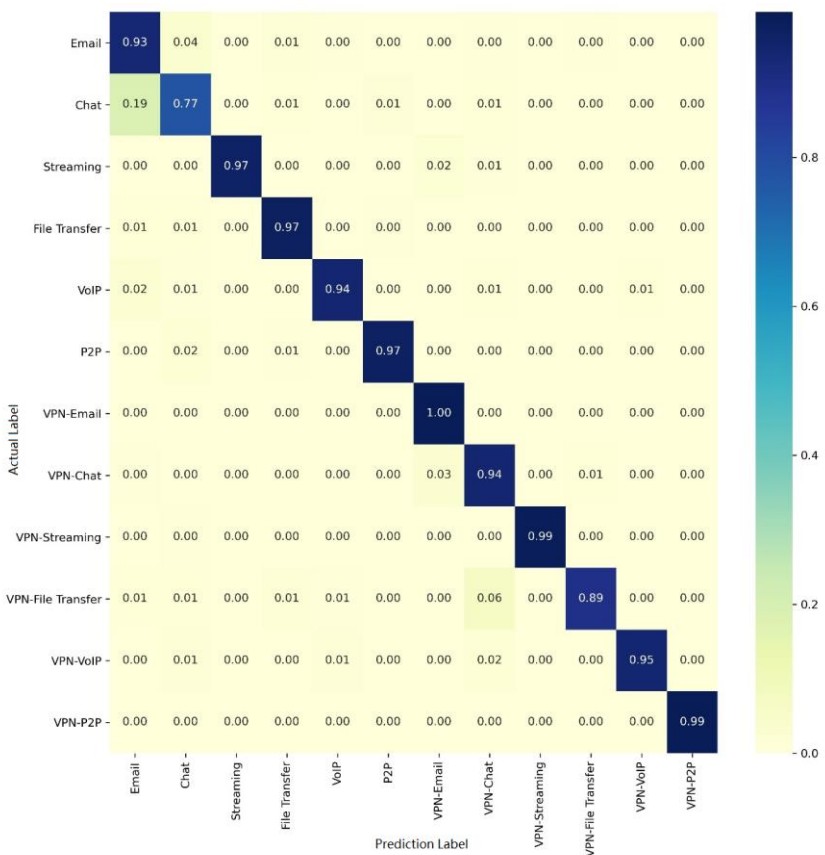

**Figure 9.** Confusion matrix for the CNN model.

The confusion matrix generated by the C-LSTM model, as depicted in Figure 10, allows us to observe the classification results. In this matrix, each row represents the traffic type, while each column corresponds to the predicted traffic type of the C-LSTM model. It is worth noting that Chat-type traffic is misclassified as Email-type due to similarities in traffic characteristics or the presence of similar application types within the actual traffic. This poses challenges in accurately identifying carried traffic solely based on analysis. To further enhance model accuracy, we can consider incorporating additional information beyond individual packet analysis for multimodality-based traffic identification.

During the training of both the CNN model and the C-LSTM model, we closely monitored the trend of the cross-entropy loss function values with each batch iteration. As depicted in Figure 11, both models exhibited a gradual reduction in their loss function values, indicating continuous learning and performance improvement. However, it is worth noting that the C-LSTM model displayed a notably faster rate of convergence in the reduction in the loss function. In comparison with the CNN model, it appeared to converge to a lower loss value more swiftly. This observation suggests that the C-LSTM model may possess certain advantages in leveraging sequential information and long-term dependencies, resulting in quicker learning and adaptability.

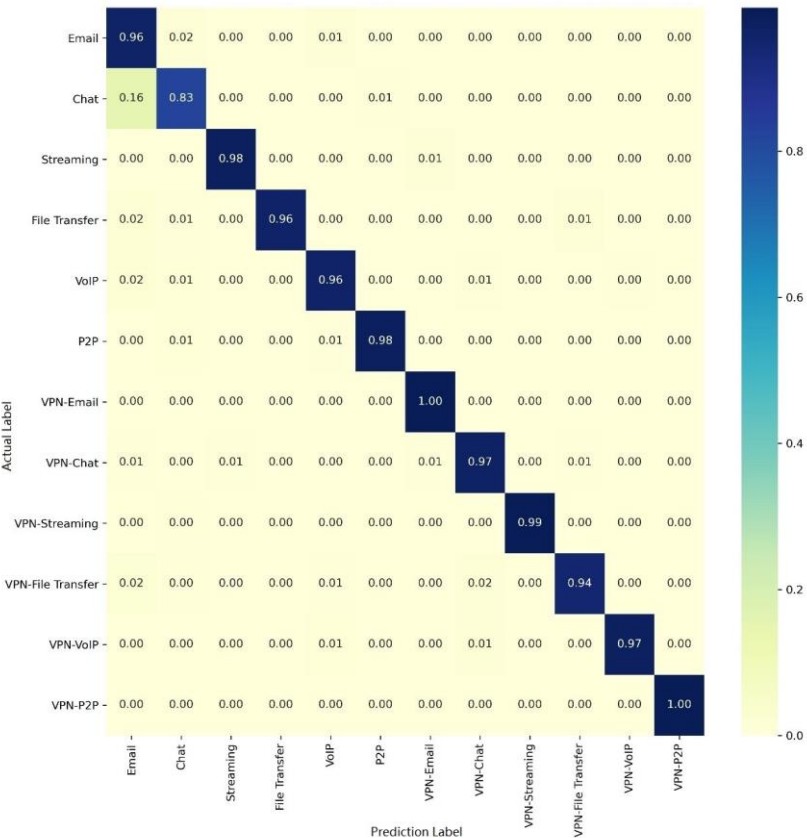

**Figure 10.** Confusion matrix for the C-LSTM model.

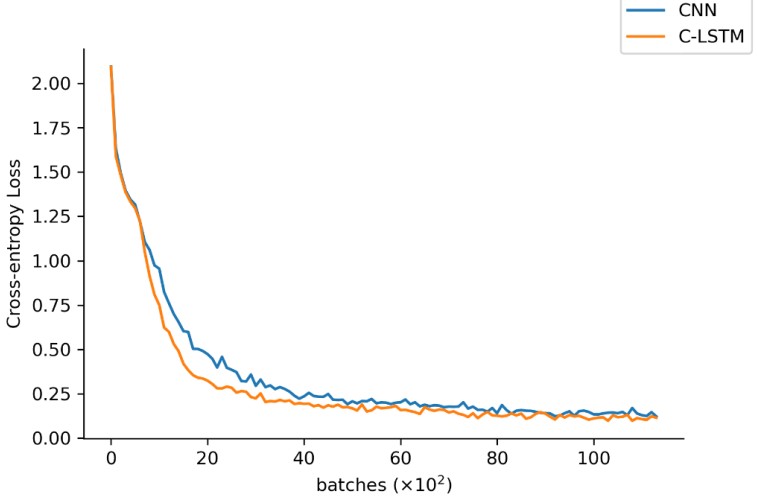

**Figure 11.** The computational efficiency of models.

The proposed CNN model in this study achieved a commendable level of precision and recall, highlighting its exceptional performance in accurately classifying and detecting intricate patterns within the dataset. Moreover, it outperforms traditional machine learning algorithms such as the C4.5 algorithm and the KNN algorithm, which rely on manually selected data features. Furthermore, the performance of the C-LSTM model was found to be superior to that of a single CNN model, as depicted in Figure 12. It shows better performance on the file transfer type and VPN-encrypted traffic type, which validates the effectiveness of the LSTM method in extracting packet temporal features.

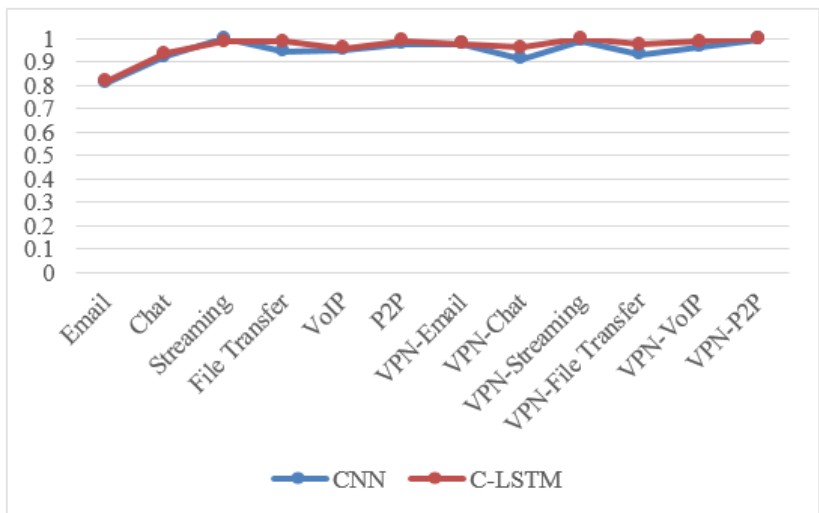

**Figure 12.** The accuracy comparison between CNN and C-LSTM models.

We present the hyperparameters used in the two deep learning models in Figure 12. After training and testing on an open-source dataset, the accuracy rate achieved was 96.4%. To verify the effectiveness of the C-LSTM model developed in this study for extracting spatial and temporal features of data packets, we conducted a quantitative comparison with other methods.

## 5. Conclusions

In this study, we delved deeply into the identification of encrypted traffic on the Internet. We observed that while encrypted communication enhances user privacy protection, it also presents challenges in traffic identification. Although mature traffic identification techniques have performed well both domestically and internationally, including deep packet inspection and traditional machine learning methods, they exhibit certain limitations in effectively identifying encrypted traffic. To address this issue, we propose an encrypted traffic identification method based on C-LSTM that can automatically preprocess traffic data and effectively address the inherent issues in traditional methods. Initially, we constructed recognition models based on CNN and achieved remarkable results. Subsequently, we refined the model and developed a model based on C-LSTM. Our experiments demonstrate that our proposed system significantly enhances the accuracy and efficiency of encrypted traffic identification by effectively extracting spatial and temporal features from encrypted traffic packets. In future research, we plan to further partition original traffic streams or sessions to achieve higher granularity with the potential integration of a real-time traffic collection assessment system's generalization capability.

**Funding:** This research received no external funding.

**Institutional Review Board Statement:** Not applicable.

**Informed Consent Statement:** Not applicable.

**Data Availability Statement:** The data presented in this study are openly available in the VPN-nonVPN dataset (ISCXVPN2016) at https://www.unb.ca/cic/datasets/vpn.html (accessed on 19 February 2016), reference number [22].

**Conflicts of Interest:** The author declares no conflict of interest.

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
