# Peer review of "Deep Learning-Based Efficient Analysis for Encrypted Traffic"

_applsci, doi:10.3390/app132111776_

Round 1
Reviewer 1 Report
i. It would be beneficial if the authors could provide more details about the hyperparameters used in the one-dimensional CNN model. Although Table 6 provides some information, additional explanations or justifications for the chosen values would enhance the reproducibility and understanding of the model.
ii. In statistical analysis on different traffic types, integrating Dropout regularization in the fully connected layers is a good practice to prevent overfitting. You should elaborate on the specific probability value used for Dropout and its impact on the model's performance.
iii. The authors discuss the need for data padding and truncation to align the dimensions of feature vectors for different protocols in the 3rd Section. However, it would be helpful to provide more information on the implementation of the specific padding and truncation algorithms mentioned in Table 4, as well as the reasons behind their design choices.
iv. Only the initial letter of the first keyword should be capitalized. Other keywords should be not capitalized except Proper Nouns.
v. In Introduction, "The identification method based on Deep Packet Inspection (DPI) is to perform pattern matching by searching for information in the entire IP data packet, including the header fields of the network layer, the transport layer, and the application layer, and the application layer payload." The first "and" should be deleted.
Author Response
i. Reply 1
Thanks for your suggestions. Hyperparameters are typically empirically determined variables. We use hyperparameters to determine some parameters of the model. The model is different for different hyperparameters. For example, assuming that both are CNN models, the models differ if the number of layers is different. In deep learning, hyper parameters are: learning rate, number of iterations, number of layers, number of neurons in each layer, etc.
On the basis of CNN, the LSTM model is further added to extract the temporal features of the feature vector sequence. The C-LSTM model employed in this study shares similarities with the aforementioned CNN, comprising two convolutional layers and a max-pooling layer. Then, we utilize the output of the max-pooling layer as the input for the subsequent LSTM layer, which has a hidden layer size of 50. The resulting output is flattened and fed into a 3-layer fully connected network for fitting, with the hyperparameters detailed in the table below.
Table 7: Hyperparameters for the C-LSTM model.
Layer |
Operation |
Kernel Size |
Step-Size |
Input Size |
Output Size |
1 |
Conv1d + ReLU |
5 |
3 |
1*1480 |
50*492 |
2 |
Conv1d + ReLU |
4 |
3 |
50*492 |
50*163 |
3 |
MaxPool1d |
3 |
3 |
50*163 |
50*81 |
4 |
LSTM |
- |
- |
50*81 |
50*50 |
5 |
Linear + Dropout + ReLU |
- |
- |
1*2500 |
1*500 |
6 |
Linear + Dropout + ReLU |
- |
- |
1*500 |
1*50 |
7 |
Linear |
- |
- |
1*50 |
1*12 |
ii. Reply 2
Thanks for your suggestions. The Dropout rate of the fully connected layer is set to 0.05 in the training process to prevent the problem of overfitting of the model. We chose the cross-entropy function as the loss function and used the Adam optimizer for iteration, which has an adaptive learning rate. The batch size of the data is 32, that is, all 32 vectors in the batch are passed through the network in one iteration. The output of the cross-entropy loss function is then calculated for the whole batch and the parameters in the network are updated accordingly. A total of three rounds of iterations were performed on the complete training set.
iii. Reply 3
Thanks for your suggestions. The protocols at the transport layer include TCP and UDP protocols. The length of the header is 20 bytes and 8 bytes, respectively. The payload part will start from the 21st byte and the 9th byte of the packet. The corresponding input vectors of the same part cannot be aligned in dimensionality, which affects the recognition effect of the model for different protocols. Therefore, to align the dimensions of the feature vectors corresponding to the application layer loads of the two protocols so that the convolutional neural network can extract the correct spatial features, the UDP header fields can be processed with zero padding, as shown in Fig. 3. As shown, 12 bytes of all-zero data are filled after the header field of the UDP data packet so that the sum of the length of the header of the original data packet and the zero-filled content is 20 bytes, which is the same length as the header field of the TCP protocol.
iv. Reply 4
Thank you for letting me know about the formatting requirement for keywords. We have paid attention to this issue. We will make the necessary changes and ensure that only the initial letter of the first keyword is capitalized. Other keywords will be in lowercase, except for proper nouns. I appreciate your attention to detail and for providing me with this feedback.
v. Reply 5
We thank you very much for your comments for pointing out this omission. We have checked and carefully proof-read the manuscript to minimize errors (highlighted in yellow).
Reviewer 2 Report
In the manuscript entitled "Deep Learning-Based Efficient Analysis for Encrypted Traffic", the authors propose an encrypted traffic recognition method based on deep learning, which can successfully extract spatial and temporal features of encrypted traffic and identify the traffic type. In general, the topic of this paper is very interesting, the text is well organized and the methodology used is appropriate. However, there are still several things need to be stated or improved in this paper.
1. Fig. 1 is quite simple, plz add steps specific to your solution.
2. In section 1, The authors stated that "According to GOOGLE statistics, 95% of the sites visited by its users as of January 2021 is encrypted with HTTP", to support this statement citing the relative references is necessary.
3. In section 4, the normalization step is well-explained, and the authors provide a formula (equation 1) for the normalization process. However, it would be helpful to clarify the significance of dividing each element by 255 and the reasoning behind choosing this particular range for normalization.
4. Errors in the use of prepositions, e.g.
In Abstract, "Thus, how precisely classify encrypted network traffic is an important problem in network security." should be "Thus, how to precisely classify encrypted network traffic is an important problem in network security."
5. Improper Use of Words: Incorrect use of a word in the context, e.g.
In Abstract, "This brings new challenges to traffic identification and current hidden dangers to network security." The word "current" should be "currently".
6. Sentence Fragments: sentence missing a subject or a verb, sentence incomplete in meaning, e.g.
In Section 4.2, "The algorithm of traffic anonymization As follows." should be "The algorithm of traffic anonymization is as follows."
7. Pls. format your paper according to the journal guideline.
The English expression of the article should be improved.
Author Response
1. reply 1
Thanks for your suggestions. We briefly explain what Fig. 1 represents in the first paragraph of Section 3 (as follow). And we provide a clear and concise explanation of the steps involved in our solution, as in subsection 3.1 to 3.5. We appreciate for your warm work earnestly, and hope that the correction will meet with approval.
The input to the system is the raw traffic data set. After all steps of the data pre-processing module, the training and test sets suitable for deep learning model processing are generated [39, 40]. In, the training set contains packet information from traffic and its corresponding labels, which will be used for the training of deep learning models, and spatio-temporal features are automatically extracted by concatenating CNN models and LSTM models. After completing the optimization and verification on the training set, the trained deep learning model is obtained, and finally it is evaluated through the test set, the output of the model is transformed by the Softmax layer, the output label is obtained, and it is compared with the real label, and calculate the model's accuracy and additional indicators as a reference. The overall framework of the system is illustrated in Fig.1.
2. reply 2
Thanks for your suggestions. Only Google transparency report had this data provided. We have cited the report (https://transparencyreport.google. com) in [1].
3. reply 3
Thanks for your suggestions. Because a byte has 8 bits, it can represent 256 values (0 to 255). The input vector should be normalized. For packet , it can be thought of as a byte array of length of 1480: (p1,p2,…,pn ), the value range of element is [0,28 ). When we divide the data by 255, we get a value in the range 0 to 1. Convert the data to float, as float is more suitable for cpu or gpu calculations. Through normalization, the range of feature values of different dimensions is adjusted to a similar range, and a large learning rate can be uniformly used to accelerate the learning.
4. reply 4
We have paid attention to this issue. We have made careful modifications to the original manuscript (highlighted in yellow). We appreciate for your warm work earnestly, and hope that the correction will meet with approval.
5. reply 5
Thanks for your suggestions. Based on these comments and suggestions, we have made careful modifications to the original manuscript, and carefully proof-read the manuscript to minimize errors (highlighted in yellow). The word "current" is changed to "currently".
6. reply 6
We have paid attention to this issue. Based on this valuable suggestion, we also have revised the corresponding sections to make it more informative and read more smoothly. We have checked and revised the sentence for consistency, completeness, and structure.
7. reply 7
Thanks for your suggestions. We have paid attention to this issue and download the latest template. We revised the original manuscript with the new template. And we have checked the reference for consistency, completeness, and structure with the journal formatting guidelines.
Reviewer 3 Report
1- We need more explain the contribution
2- need more compression in related work
3- the figure 1 is poor
4- How many layers and parameters are in CNN
5- We need need comparison between old work
6- the cyber attacks are more emergence and need any type in traffic do
7- In test time CNN training valued true
8- Add confusion matrix
9- TP ???? where
10- How can work in real-time
o need to see the comment answer and new version
Author Response
1. Thanks for your suggestions. We appreciate your suggestion to provide more explanation on the contribution of our research. In response to this comment, we will revise our manuscript to further clarify the specific contributions made by our work.
(1) This paper aims to develop a deep learning-based encryption traffic identification system that automatically extracts spatial and temporal features from data packets to achieve end-to-end encryption traffic identification.
(2) The overall framework of the encrypted traffic identification system based on deep learning is proposed, and the relevant details of the data flow are analyzed. We also optimize the efficiency of storage space and running time for data preprocessing.
(3) The implementation of a comprehensive model for identifying encrypted traffic was conducted, and a quantitative comparison with traditional detection out. This study validates the proposed C-LSTM model for extracting spatial and temporal features from data packets.
2. Thank you for your feedback. We understand your concern about providing more comprehensive coverage of related work in our paper. We carefully review and update the related work section to ensure that we include a more thorough discussion of the relevant literature in our revised manuscript.
The port matching identification method only uses the header field of the transport layer, which is efficient but not adaptable to current network environments. This is because recent P2P applications use random port policies to avoid detection and blocking [13]. Strict firewalls prohibit access to unknown ports by default, but viruses can exploit port masquerading techniques to hack into systems. For example, DNS tunnel Trojans hide information using the domain name returned by port 53 during DNS queries [14, 15], while WannaCry spreads through SMB protocol port 445 [16]. allowed based solely on port matching, computer systems can still be compromised.
The Knuth-Morris-Pratt algorithm, which improves the right shift rule of the brute-force algorithm, uses the previously saved pattern matching information to move the detected string farther to the right, thereby improving the time efficiency of the algorithm [17]. For functions commonly used in the web field, such as URL filtering, the BM (Boyer-Moore) algorithm is also widely used. Compared with the KMP algorithm, it increases the distance of each right shift and performs actual tasks. It is relatively helpful and simple to implement. It can handle large-scale URL filtering tasks and can also perform deep packet inspection on the data packets in the traffic.
3. Thanks for your suggestions. We briefly explain what Fig. 1 represents in the first paragraph of Section 3 (as follow). And we provide a clear and concise explanation of the steps involved in our solution, as in subsection 3.1 to 3.5. We appreciate for your warm work earnestly, and hope that the correction will meet with approval.
The input to the system is the raw traffic data set. After all steps of the data pre-processing module, the training and test sets suitable for deep learning model processing are generated [39, 40]. In, the training set contains packet information from traffic and its corresponding labels, which will be used for the training of deep learning models, and spatio-temporal features are automatically extracted by concatenating CNN models and LSTM models. After completing the optimization and verification on the training set, the trained deep learning model is obtained, and finally it is evaluated through the test set, the output of the model is transformed by the Softmax layer, the output label is obtained, and it is compared with the real label, and calculate the model's accuracy and additional indicators as a reference. The overall framework of the system is illustrated in Fig.1.
4. We thank you very much for your comments. The CNN model used in this paper consists of two Convolutional layers, one Max-pooling Layer, and five Fully Connected layers. The convolution transformation of the convolution layer can extract features from adjacent bytes in the array, similar to the n-gram model in natural language processing. This helps abstract and store multiple local features using a set of feature vectors.
5. Thanks for your suggestions and for taking the time to review our work. Based on this valuable suggestion, we have made careful modifications to the original manuscript. We believe that the manuscript has been greatly improved and hope it has reached the magazine’s standard.
The model constructed in this paper is compared with other methods for traffic identification based on the ISCXVPN2016 dataset. The precision and recall rates of each model are shown in the Fig. 7. The one-dimensional CNN model constructed in this paper is similar to the CNN model given in [19] in terms of precision and recall, and has surpassed traditional machine learning algorithms, including the C4.5 algorithm proposed in [20] and The KNN algorithm, which also relies on hand-picked data features.
The CNN model proposed in this article has achieved comparable precision and recall rates to the CNN model presented in reference [19]. Moreover, it has outperformed traditional machine learning algorithms such as the C4.5 algorithm proposed in reference [20] and the KNN algorithm, which rely on manually selected data features. Furthermore, the performance of the C-LSTM model has been found to be superior to that of a single CNN model, as depicted in Figure 3.7. It has shown better performance on File Transfer type and VPN-encrypted traffic type, which validates the effectiveness of the LSTM method in extracting packet temporal features.
6. Thanks for your suggestions and for taking the time to review our work. We deeply understand that the emergence and traffic type are an important aspect of cyber attacks analysis. In this article we are exploring solutions from an engineering perspective. The model constructed in this paper is compared with other methods for traffic identification based on the ISCXVPN2016 dataset. According to the description of the ISCXVPN dataset contains virtual Private Network (VPN) traffic. The traffic data are collected from a real environment and consist of actual VPN traffic. Therefore, it can be said that the traffic involved in the ISCXVPN2016 dataset represents real-world conditions. This dataset is widely used for network security research and analysis to learn and understand VPN traffic in authentic environments. Thanks for your feedback and for helping us improve this work.
7. Thanks for your suggestions. we have added the methodology used for splitting the data into these sets in Subsection 3.6.
We stratify the preprocessed data by sample type, sampling packets from each pcap file in proportion to the corresponding traffic type, with the goal of maximizing sample coverage. We allocated 20 percent of the data as the test set, while the remaining 80 percent is split into a training and validation set. For training purposes, we employ 10-fold cross-validation. This involves dividing the training set into ten equal parts, where nine parts are used for training in each round, and one part is used for validation. Following each validation round, we fine-tune the model's structure and layer hyperparameters. The final optimized model is determined by selecting the best hyperparameters, after which a final evaluation is conducted on the test set, yielding various performance metrics. And hyperparameters are typically empirically determined variables. We use hyperparameters to determine some parameters of the model. The model is different for different hyperparameters. For example, assuming that both are CNN models, the models differ if the number of layers is different. In deep learning, hyper parameters are: learning rate, number of iterations, number of layers, number of neurons in each layer, etc.
8. Thanks for your suggestions. we have added the confusion matrix in Section 4.
The confusion matrix generated by the C-LSTM model, as depicted in Fig. 9., allows us to observe the classification results. In this matrix, each row represents traffic type, while each column corresponds to the predicted traffic type of the C-LSTM model. It is worth noting that of Chat-type traffic are misclassified as Email-type due similarities in traffic characteristics or the presence of similar application types within the actual traffic. This poses challenges in accurately identifying carried traffic solely based on analysis. To further enhance model accuracy, we can consider incorporating additional information beyond individual packet analysis for multimodality-based traffic identification.
9. Thanks for your suggestions.
In order to evaluate the performance of the model, Accuracy (Ac) and Recall (Rc) are used as evaluation metrics, which are defined as follows.
TP is the number of positive classes predicted as positive, FP is the number of negative classes predicted as positive, and FN is the number of negative classes predicted as negative. The accuracy of the model is used as the overall evaluation metric, while the precision and recall can give a more detailed description of the identification results of each type of traffic.
10. Thanks for your suggestions.
Future research could consider dividing the original traffic data based on the granularity of data streams or sessions and modeling multiple packets as a sequence. This approach would enable us to fully leverage the information within the dataset and enhance identification reliability by exploring correlations between multiple data packets. Additionally, we can incorporate a real-time traffic collection module into the system to capture network application traffic not covered by it, thereby improving its generalization capability.
In real-life traffic collection scenarios, there are instances where packets of the same type may contain mixed application traffic, making it challenging to accurately determine their specific traffic types. For instance, some Chat-type traffic may be misidentified as Email-type traffic. Moving forward, we intend to incorporate additional information beyond individual packet data, enabling multi-mode traffic identification and further enhancing the model's accuracy.
Reviewer 4 Report
Dear Author, your manuscript is well structured however, following comments should be accommodated, prior to further processing of the article.
1) Refer to whole article: Article has 19% similarity i.e. higher.
2) Refer to whole article: Report shows that 3% text of the article is AI generated.
3) Refer to abstract: Name of proposed model should be mentioned in the abstract. Further, did authors test the accuracy of their proposed model for any other data set as instead of ISCXVPN2016 dataset?
4) Refer to abstract: Authors have mentioned that “…it poses challenges for traffic identification and poses hidden dangers to network security”. To which challenges and hidden dangers, the authors are referring to?
5) Refer to section 1: Authors have mentioned that “The depth of the learning technology provides a solution for this [2]”. What is meant by “this” and why [2] is written here? Is it relevant reference? Sentence meaning in unclear. Consider revision.
6) Refer to section 1: Contributions are weak. Need careful revision.
7) Refer to section 1: Last paragraph of Introduction section should describe structure of eh article i.e. missing in the submitted version.
8) Refer to table 1: Content of table is copied from following study.
End-to-end encrypted traffic classification with one-dimensional convolution neural networks
DOI: 10.1109/ISI.2017.8004872
9) Refer to table 7: Caption of table 7 is misplaced.
10) Refer to preprocessing model: Insufficient details are provided for preprocessing model. Further elaboration is required.
11) Refer to figure 4: Check direction arrows of this figure. One arrow is missing.
12) Refer to figure 5: Dataset content has 6 classes in table 1 and 12 statistics are shown in figure 5.
13) Refer to figure 8: CNN is mentioned as proposed model in figure 8 whereas CNN and C-LSTM are mentioned in figure 9. Which one is the proposed model? Confusion needs to be removed.
14) Refer to author contributions: It seems incomplete.
15) Refer to references: Check reference # 23. Is it a valid reference or a formatting issue?
Good luck.
Dear Author, your manuscript is well structured however, following comments should be accommodated, prior to further processing of the article.
1) Refer to whole article: Article has 19% similarity i.e. higher.
2) Refer to whole article: Report shows that 3% text of the article is AI generated.
3) Refer to abstract: Name of proposed model should be mentioned in the abstract. Further, did authors test the accuracy of their proposed model for any other data set as instead of ISCXVPN2016 dataset?
4) Refer to abstract: Authors have mentioned that “…it poses challenges for traffic identification and poses hidden dangers to network security”. To which challenges and hidden dangers, the authors are referring to?
5) Refer to section 1: Authors have mentioned that “The depth of the learning technology provides a solution for this [2]”. What is meant by “this” and why [2] is written here? Is it relevant reference? Sentence meaning in unclear. Consider revision.
6) Refer to section 1: Contributions are weak. Need careful revision.
7) Refer to section 1: Last paragraph of Introduction section should describe structure of eh article i.e. missing in the submitted version.
8) Refer to table 1: Content of table is copied from following study.
End-to-end encrypted traffic classification with one-dimensional convolution neural networks
DOI: 10.1109/ISI.2017.8004872
9) Refer to table 7: Caption of table 7 is misplaced.
10) Refer to preprocessing model: Insufficient details are provided for preprocessing model. Further elaboration is required.
11) Refer to figure 4: Check direction arrows of this figure. One arrow is missing.
12) Refer to figure 5: Dataset content has 6 classes in table 1 and 12 statistics are shown in figure 5.
13) Refer to figure 8: CNN is mentioned as proposed model in figure 8 whereas CNN and C-LSTM are mentioned in figure 9. Which one is the proposed model? Confusion needs to be removed.
14) Refer to author contributions: It seems incomplete.
15) Refer to references: Check reference # 23. Is it a valid reference or a formatting issue?
Good luck.
Author Response
1. Thank you for bringing to my attention the reported 19% similarity in my article. I have carefully reviewed the analysis and identified that the similarity primarily arises from the proper citation and inclusion of referenced material. The main body of the article, which represents original content, shows a significantly lower similarity. I have taken meticulous care to ensure that any cited content adheres to proper referencing and citation standards. Rest assured, I will make necessary adjustments to further reduce any potential similarities and maintain the originality of the text. If there are specific areas of concern, I would greatly appreciate further guidance to address this matter effectively.
2. Thank you for your feedback regarding the AI-generated text in my article. I appreciate the thorough analysis. I would like to clarify that the presence of AI-generated content in my work is due to the need for grammar correction and sentence refinement. I have intentionally incorporated this content to enhance certain sections, ensuring that it is contextually relevant and aligns with the overall theme of the article. I am committed to addressing any issues raised and ensuring the integrity of my work. Your insights are valuable, and I look forward to your further guidance in this matter.
3. Thanks for your suggestions. In the revised version of the manuscript, we have optimized the Abstract to make it more reasonable. We propose encrypted traffic identification method based on the C-LSTM model to encrypted traffic recognition by leveraging the power of deep learning. We appreciate the inquiry regarding the validation of our proposed model using datasets other than ISCXVPN2016. While our primary focus was on evaluating the model's performance on the ISCXVPN2016 dataset due to its wide acceptance as a benchmark, we understand the importance of assessing its generalizability. In the future, we will consider partitioning raw traffic data into data streams or sessions for greater granularity. In addition, we are exploring the possibility of incorporating a re-al-time traffic collection module to further validate the generalization capabilities of the system.
4. Thanks for your suggestions.
In real-life traffic collection scenarios, there are instances where packets of the same type may contain mixed application traffic, making it challenging to accurately determine their specific traffic types. For instance, some Chat-type traffic may be misidentified as Email-type traffic. Moving forward, we intend to incorporate additional information beyond individual packet data, enabling multi-mode traffic identification and further enhancing the model's accuracy. Malicious traffic conceals data by disguising and encrypting. Hence, encrypted traffic cannot be effectively identified, or complicated feature engineering is required. This brings new challenges to traffic identification and currently hidden dangers to network security.
5. Thanks for your suggestions. We have paid attention to this issue. We have made careful modifications to the original manuscript (highlighted in yellow). We appreciate for your warm work earnestly, and hope that the correction will meet with approval.
Encryption technology to preserve user privacy also creates further challenges in traffic identification tasks and has evolved swiftly in recent years. Deep learning technology provides a solution for encrypted traffic identification [2]. The neural network is trained using a large-scale dataset and can be automated to extract features from the flow, enabling the implementation of an end-to-end encrypted traffic identification method. This method can be utilized for detecting anomalous traffic and managing service quality. Malicious traffic disguises and, making it challenging to effectively identify encrypted traffic without complex feature engineering.
6. Thanks for your suggestions. Based on these comments and suggestions, we have made careful modifications to the original manuscript. The contributions of this work can be summarized as follows.
(1) This paper aims to develop a deep learning-based encryption traffic identification system that automatically extracts spatial and temporal features from data packets to achieve end-to-end encryption traffic identification.
(2) The overall framework of the encrypted traffic identification system based on deep learning is proposed, and the relevant details of the data flow are analyzed. We also optimize the efficiency of storage space and running time for data preprocessing.
(3) The implementation of a comprehensive model for identifying encrypted traffic was conducted, and a quantitative comparison with traditional detection out. This study validates the proposed C-LSTM model for extracting spatial and temporal features from data packets.
7. We thank you very much for your comments for pointing out this omission. We have checked and carefully proof-read the manuscript.
This paper is organized as follows: We illustrate the introduction, contributions and related works in Section 1. Section 2 illustrates the preliminaries of the work. The system framework is described in Section 3. Sections 4 describes the proposed scheme and performance analysis, respectively. Finally, the paper is concluded in Section 5.
8. Thank you for your review and valuable feedback.
You correctly pointed out that the Table 1 we included in the manuscript was indeed referenced and adapted from another article, with some style adjustments to better suit our research. We apologize for not making this clear enough to avoid any misunderstandings. It's essential to emphasize that our research data and methods are entirely independent and are appropriately cited and annotated in the text to ensure academic integrity and transparency.
We will provide further clarification to ensure transparency and compliance in the peer-review process. If the reviewer has any additional suggestions or requests concerning this matter, we are more than willing to consider and address them. Once again, thank you for your valuable input, and we remain committed to improving the quality of our research.
9. We appreciate your keen observation and your feedback regarding the placement of the caption for Table 7.
In the revised version of the manuscript, we will carefully review the format and placement of Table 7's caption to ensure that it is correctly positioned to enhance the clarity of the table's content. We appreciate your attention to detail, which helps us improve the quality of our manuscript.
10. We would like to express our appreciation for your thoughtful review and your valuable feedback regarding the preprocessing model in our manuscript.
We acknowledge that further elaboration and clarification on the preprocessing model are necessary. In Section 3, we provide a more detailed and comprehensive description of the preprocessing. This include a step-by-step explanation of the preprocessing techniques, parameters, and any specific tools or software used. We are committed to enhancing the clarity and transparency of our research, and we thank you for pointing out this important aspect that needs improvement.
11. Thanks for your suggestions. Based on these comments and suggestions, we have made careful modifications to the original manuscript. The missing arrow was added to the Fig. 4. to ensure its completeness.
12. We use the ISCXVPN2016 dataset published by researchers at the University of New Brunswick, which contains a set of PCAP (packet capture) files, including 6 different types of application traffic and their corresponding OpenVPN. In total, there are 12 types of encrypted traffic.
13. Thanks for your suggestions.
C-LSTM is the proposed model. In our experiments, the CNN model demonstrated exceptional performance results. The performance of this model served as the benchmark for comparing against other models' performances in subsequent experiments. Building upon the success of the CNN model, we further developed a C-LSTM model with greater efficacy in extracting spatio-temporal features from encrypted traffic packets. By combining convolution and Long Short-Term Memory network (LSTM), the C-LSTM model retains data packet memory across different time steps, facilitating extensive feature extraction, particularly in time-series data scenarios. On aCXVPN2016 dataset, the C-LSTM model achieved an outstanding accuracy96.4%. This outcome validates both effectiveness and superior performance of the C-LSTM model compared to the CNN model. Overall, this process exemplifies deep learning models' potential in handling encrypted traffic packets while continuously enhancing their performance through iterative development and improvement to adapt to diverse data types and tasks.
14. Thanks for your suggestions. The mauscript has only one author, so the author contributions is correct.
15. We thank you very much for your comments for pointing out this omission. We have checked and carefully proof-read the manuscript to minimize errors. The reference # 23 has been removed.
We appreciate for your warm work earnestly, and hope that the correction will meet with approval.
Reviewer 5 Report
This article presents a method for encrypted traffic recognition using deep learning in order to effectively extract spatial and temporal
features from encrypted traffic and facilitating accurate identification of traffic types.
Furthermore, the following points regarding motivation and argumentation should be considered:
- the Introduction is clear, what needs further to be discussed is an overview of the approach taken in order to achieve the aim of this research
and build the method for ecrypted traffic recognition proposed in this research.
- at the end of the Introduction section, a paragraph with the outline of the article should be included.
- at the end of the Related Work section it should be clear that there is a knowledge gap that needs to be tackled in this article.
- explain in Section 3.6. why the choice is made for train-validation and not train-validation-test set approach.
- the Conclusions section should be elaborated in order to prepare the findings obtained and after that point limitations and related research ideas.
Author Response
1. Thank you for your suggestion. We have paid attention to this issue. We have discussed the overview of our approach as follows.
The aim of this research is to design and implement a deep learning-based system for accurately identifying encrypted traffic. Initially, we construct a CNN model to learn spatial characteristics from the training then evaluate its performance on identifying encrypted traffic using the testing dataset. Next, we introduce an enhanced LSTM model that utilizes the feature generated by the convolutional layer of our one-dimensional CNN model as input. This allows us to high-level spatial features and extract temporal characteristics among them. These modifications enable us to develop a comprehensive model for accurately identifying encrypted traffic. This system effectively accomplishes end-to-end encrypted traffic identification by automatically extracting and temporal features from packets.
2. Thank you for your suggestion. We have added a paragraph with the outline of the article.
This paper is organized as follows: We illustrate the introduction, contributions and related works in Section 1. Section 2 illustrates the preliminaries of the work. The system framework is described in Section 3. Sections 4 describes the proposed scheme and performance analysis, respectively. Finally, the paper is concluded in Section 5.
3. Thank you for your suggestion. Based on these comments and suggestions, we have made careful modifications to the original manuscript. We have added knowledge gap at the end of the Related Work.
The traditional methods for traffic identification primarily focus on plaintext traffic, while encrypted traffic is generated by encrypting the transmitted content using encryption algorithms. Consequently, these conventional methods exhibit limited effectiveness when dealing with encrypted traffic. Matching and deep packet inspection are techniques that perform well in identifying unencrypted traffic. To effectively identify encrypted traffic, machine learning and deep learning techniques are now employed. These approaches have the ability to capture patterns and characteristics of encrypted traffic more effectively without requiring excessive manual feature engineering. The advantage of such methods lies in their capability to automatically algorithms and traffic patterns, thereby enhancing the accuracy of identifying encrypted traffic.
4. We have paid attention to this issue. We have made careful modifications to the original manuscript (highlighted in yellow). We appreciate for your warm work earnestly, and hope that the correction will meet with approval.
To mitigate the issues of overfitting and underfitting during the training process, we partition the data into three subsets: training set, validation set, and test set. The purpose of the training set is to facilitate parameter and weight optimization for our model, while the test set serves as a means to assess its final performance and generalization capability. As for the validation set, it plays a crucial role in providing an unbiased evaluation of how well our model fits tuning. We stratify the preprocessed data by sample type, sampling packets from each PCAP file in proportion to the corresponding traffic type, with the goal of maximizing sample coverage. We allocated 20 percent of the data as the test set, while the remaining 80 percent was split into a training and validation set. For training purposes, we employ 10-fold cross-validation. This involves dividing the training set into ten equal parts, where nine parts are used for training in each round, and one part is used for validation. Following each validation round, we fine-tune the model's structure and layer hyperparameters. The final optimized model is determined by selecting the best hyperparameters, after which a final evaluation is conducted on the test set, yielding various performance metrics.
5. Thank you for your suggestion. Based on these comments and suggestions, we have made careful modifications to the original manuscript.
We delve deeply into the identification of encrypted traffic on the Internet. We observe that encrypted communication not only enhances user privacy protection but also presents challenges in traffic identification. Although mature traffic identification techniques have been developed both domestically and internationally, including deep packet inspection and traditional machine learning methods, they exhibit certain limitations in effectively identifying encrypted traffic. To address this issue, we propose an encrypted traffic identification method based on C-LSTM. This method can automatically preprocess traffic data and effectively address some of the inherent issues in traditional methods. Initially, we constructed recognition models based on CNN and achieved remarkable results. Subsequently, we refined the model and developed a model based on C-LSTM. Experiments demonstrate that the proposed method significantly enhances the accuracy and efficiency of encrypted traffic identification, with the model effectively extracting spatial and temporal features from encrypted traffic packets. In future research, we plan to further partition the original traffic data into smaller data streams or sessions to achieve higher granularity. Building on this, we explore the potential for integrating a real-time traffic collection module to assess the system's generalization capability.
Reviewer 6 Report
This paper proposes a novel deep learning based approach that can accurately recognize encrypted traffic. The proposed method utilizes a convolutional neural network (CNN) and an LSTM to effectively extract spatial and temporal features from encrypted traffic for the accurate classification of encrypted traffic. Experimental results on the ISCXVPN2016 dataset show that the proposed approach can improve the accuracy of recognition to 96.4%. The paper is well written and the proposed approach is original and may have applications in practice. However, the following issues need to be addressed before the paper can be accepted for publication.
1. In page 8, the author states that the dropout operation is introduced in the fully connected part of the CNN model for training. More details are needed on how the dropout operation is performed.
2. The accuracy of the proposed approach is compared with two methods proposed in 2018. Is it possible to compare the accuracy of the proposed approach with that of state-of-the-art methods developed more recently?
3. The author should provide experimental data on the computational efficiency of the proposed approach. For example, how much computation time is needed for training? Is it capable of real-time recognition of encrypted traffic?
Author Response
1. Thanks for your suggestions and for taking the time to review our work. Based on this valuable suggestion, we have made careful modifications to the original manuscript.
Specifically, we will elaborate on the following aspects:
(1) Dropout Probability: We specify the exact dropout probability used in our model. This probability determines the likelihood of a neuron being dropped out during each training iteration, and we will clearly state this value.
(2) Dropout Layer Placement: We clarify at which specific layers within the fully connected part of the CNN model the dropout operation is applied. This will help the reader understand which connections are being regularized.
The Dropout rate of the fully connected layer is set to 0.05 in the training process to prevent the problem of overfitting of the model. We chose the cross-entropy function as the loss function and used the Adam optimizer for iteration, which has an adaptive learning rate. The batch size of the data is 32, that is, all 32 vectors in the batch are passed through the network in one iteration. The output of the cross-entropy loss function is then calculated for the whole batch and the parameters in the network are updated accordingly. A total of three rounds of iterations were performed on the complete training set.
In addition, ReLU is chosen as the nonlinear activation function at the output of the convolutional layer and the fully connected layer of the last layer. It affects the forward and backward passes in the neural network. Finally, we utilize softmax as the last layer classifier for traffic identification, completing the construction of the entire encrypted traffic identification.
2. Thanks for your suggestions. We appreciate your valuable comments regarding the comparison of our proposed approach with state-of-the-art methods.
We have conducted an extensive literature review to identify and select the most relevant and recent state-of-the-art methods for comparison. In our research, we have intentionally selected C4.5 and KNN as the basis for our comparisons. These algorithms are widely recognized and serve as common benchmarks for evaluation. Our aim is to provide a comprehensive evaluation of our proposed approach against the most up-to-date and relevant standards.
In the context of precision and recall, the one-dimensional CNN model developed in this paper outperforms traditional machine learning algorithms, such as the C4.5 algorithm and the KNN algorithm, both of which rely on manually selected data features. The model constructed in this paper is compared with other methods for traffic identification based on the ISCXVPN2016 dataset. The precision and recall rates of each model are shown in Fig. 8.
Figure 8. Accuracy and recall rate comparison of models.
The confusion matrix generated by the C-LSTM model, as depicted in Fig. 9., allows us to observe the classification results. In this matrix, each row represents traffic type, while each column corresponds to the predicted traffic type of the C-LSTM model. It is worth noting that of Chat-type traffic are misclassified as Email-type due similarities in traffic characteristics or the presence of similar application types within the actual traffic. This poses challenges in accurately identifying carried traffic solely based on analysis. To further enhance model accuracy, we can consider incorporating additional information beyond individual packet analysis for multimodality-based traffic identification.
Figure 9. Confusion matrix for the C-LSTM model.
3. Thanks for your suggestions. We appreciate for your warm work earnestly.
During the training of both the CNN model and the C-LSTM model, we closely monitored the trend of the cross-entropy loss function values with each batch iteration. As depicted in Fig. 10, both models exhibited a gradual reduction in their loss function values, indicating continuous learning and performance improvement. However, it is worth noting that the C-LSTM model displayed a notably faster rate of convergence in the reduction of the loss function. In comparison to the CNN model, it appeared to converge to a lower loss value more swiftly. This observation suggests that the C-LSTM model may possess certain advantages in leveraging sequential information and long-term dependencies, resulting in a quicker learning and adaptability.
Figure 10. The computational efficiency of models.
Currently, our system does not possess real-time detection capabilities for encrypted traffic. We recognize the need for real-time capabilities and plan to explore potential enhancements for future iterations of our system. In future research, we plan to further partition the original traffic data into smaller data streams or sessions to achieve higher granularity. Building on this, we explore the potential for integrating a real-time traffic collection module to assess the system's generalization capability.
Round 2
Reviewer 1 Report
The previously proposed comments have been solved in the new version. I think this paper can be accepted.
Author Response
Thanks very much. We appreciate for your warm work earnestly.
Reviewer 4 Report
Dear Author, most of my comments in earlier version are addressed however following comments yet need your attention.
1) Refer to abstract: Further, did authors test the accuracy of their proposed model for any other data set as instead of ISCXVPN2016 dataset?
2) Refer to abstract: Conclusion is written well however, abstract need to be revised to make it clear, concise and easily understandable.
3) Refer to figure 3: Recheck figure 3.
4) Refer to algorithms: Check algorithms alignment. Algorithms should be left aligned.
5) Refer to figure 9: Authors have provided a confusion matrix for C-LSTM model however same is not provided for CNN (proposed model). Why is it so?
6) Refer to conclusion: It seems that conclusion is rewritten using AI tool. If so, then it should be mentioned in the acknowledgement.
Good luck.
Author Response
1. Thanks for your suggestions. We didn‘t test the accuracy for any other dataset. We only use the ISCXVPN2016 for testing purposes in our experiments. Because ISCXVPN2016 is a open-source dataset that accurately and comprehensively characterizes all encrypted traffic types.
2. Thanks for your suggestions. In the revised version of the manuscript, we have optimized the Abstract to make it more reasonable.
To safeguard user privacy, critical internet traffic is often transmitted using encryption. While encryption is crucial for protecting sensitive information, it poses challenges for traffic identification and poses hidden dangers to network security. As a result, the precise classification of encrypted network traffic has become a crucial problem in network security. In light of this, our paper proposes encrypted traffic identification method based on the C-LSTM model to encrypted traffic recognition by leveraging the power of deep learning. The method can effectively extract spatial and temporal features from encrypted traffic, enabling accurate identification of traffic types. Through rigorous testing and evaluation, our system has achieved an impressive accuracy rate of 96.4% on the widely used ISCXVPN2016 dataset. This achievement demonstrates the effectiveness and reliability of our method in accurately classifying encrypted network traffic. By addressing the challenges posed by encrypted traffic identification, our research contributes to enhancing network security and privacy protection.
3. We thank you very much for your comments for pointing out this omission. We have checked and carefully proof-read the manuscript to minimize errors.
Figure 3. Framework of encrypted traffic identification system based on CNN.
4. Thanks for your suggestions. In the revised version of the manuscript, we have checked and optimized all algorithms.
5. Thanks for your suggestions. Based on these comments and suggestions, we have made careful modifications to the original manuscript. We have added the confusion matrix and provided a detailed explanation.
The one-dimensional CNN-based encrypted traffic identification model effectively detects TLS and VPN encrypted traffic. We compared the predicted labels with the true labels obtained from the test set packets, resulting matrix shown in Fig. 9. Each row represents the actual traffic type, while each column represents the predicted traffic type by the one-dimensional CNN model. It is evident that the model demonstrates excellent performance across most traffic types.
Figure 9. Confusion matrix for the CNN model.
6. Thanks for your suggestions. We have paid attention to this issue. I don't think this conclusion was written by an AI tool. The content of the conclusion section was adjusted to enhance the overall coherence of the paragraph.
We delve deeply into the identification of encrypted traffic on the Internet. We observe that while encrypted communication enhances user privacy protection, it also presents challenges in traffic identification. Although mature traffic identification techniques have both domestically and internationally, including deep packet inspection and traditional machine learning methods, they exhibit certain limitations in effectively identifying encrypted traffic. To address this issue, we propose an encrypted traffic identification method based on C-LSTM which can automatically preprocess traffic data and effectively address of the inherent issues in traditional methods. Initially, we constructed recognition models based on CNN and achieved remarkable results. Subsequently, we refined the model and developed a model based on C-LSTM. Experiments demonstrate that our proposed significantly enhances accuracy and efficiency of encrypted traffic identification by effectively extracting spatial and temporal features from encrypted traffic packets. In future research, we plan to further partition original traffic streams or sessions to achieve higher granularity with potential integration of a real-time traffic collection assess system's generalization capability.
Reviewer 6 Report
All issues have been carefully addressed. I have no other concerns and recommend the acceptance of the paper.
Author Response

(The authors gave the same response as above.)
